# RNA editing enzyme APOBEC3A promotes pro-inflammatory M1 macrophage polarization

Emad Y. Alqassim[1,2], Shraddha Sharma[3,14,17], A. N. M. Nazmul H. Khan[4,17], Tiffany R. Emmons[5], Eduardo Cortes Gomez [6], Abdulrahman Alahmari [7,8], Kelly L. Singel[5,15], Jaron Mark[9,16], Bruce A. Davidson [10], A. J. Robert McGray[11], Qian Liu[6], Brian D. Lichty[12], Kirsten B. Moysich[1], Jianmin Wang [6], Kunle Odunsi[5,9,11], Brahm H. Segal [4,5,13,18✉] & Bora E. Baysal [3,18✉]

Pro-inflammatory M1 macrophage polarization is associated with microbicidal and antitumor responses. We recently described APOBEC3A-mediated cytosine-to-uracil (C > U) RNA editing during M1 polarization. However, the functional significance of this editing is unknown. Here we find that APOBEC3A-mediated cellular RNA editing can also be induced by influenza or Maraba virus infections in normal human macrophages, and by interferons in tumor-associated macrophages. Gene knockdown and RNA_Seq analyses show that APO-BEC3A mediates C>U RNA editing of 209 exonic/UTR sites in 203 genes during M1 polarization. The highest level of nonsynonymous RNA editing alters a highly-conserved amino acid in *THOC5*, which encodes a nuclear mRNA export protein implicated in M-CSF-driven macrophage differentiation. Knockdown of APOBEC3A reduces *IL6*, *IL23A* and *IL12B* gene expression, CD86 surface protein expression, and TNF-α, IL-1β and IL-6 cytokine secretion, and increases glycolysis. These results show a key role of APOBEC3A cytidine deaminase in transcriptomic and functional polarization of M1 macrophages.

[1] Department of Cancer Prevention and Control, Roswell Park Comprehensive Cancer Center, Buffalo, NY 14203, USA. [2] Department of Pathology, Faculty of Medicine, Jazan University, Jazan 45142, Saudi Arabia. [3] Department of Pathology, Roswell Park Comprehensive Cancer Center, Buffalo, NY 14203, USA. [4] Department of Internal Medicine,, Roswell Park Comprehensive Cancer Center, Buffalo, NY 14203, USA. [5] Department of Immunology,, Roswell Park Comprehensive Cancer Center, Buffalo, NY 14203, USA. [6] Department of Biostatistics/Bioinformatics, Roswell Park Comprehensive Cancer Center, Buffalo, NY 14203, USA. [7] Department of Pharmacology and Therapeutics, Roswell Park Comprehensive Cancer Center, Buffalo, NY 14203, USA. [8] Department of Medical Laboratory Sciences, Prince Sattam Bin Abdulaziz University, Al-Kharj 16278, Saudi Arabia. [9] Department of Gynecologic Oncology,, Roswell Park Comprehensive Cancer Center, Buffalo, NY 14203, USA. [10] Departments of Anesthesiology, Pathology and Anatomical Sciences, Jacobs School of Medicine and Biomedical Sciences, University at Buffalo, Buffalo, NY 14203, USA. [11] Center for Immunotherapy, Roswell Park Comprehensive Cancer Center, Buffalo, NY 14203, USA. [12] McMaster Immunology Research Centre, McMaster University, 1200 Main St W, Hamilton, ON L8N 3Z5, Canada. [13] Departments of Medicine, Jacobs School of Medicine and Biomedical Sciences, University at Buffalo, Buffalo, NY 14203, USA. [14]Present address: Translate Bio, Lexington, MA 02421, USA. [15]Present address: Office of Evaluation, Performance, and Reporting, National Institutes of Health, Bethesda, MD 20892, USA. [16]Present address: The Start Center for Cancer Care, 4383 Medical Drive, San Antonio, TX 78229, USA. [17]These authors contributed equally: Shraddha Sharma, A.N.M. N.H. Khan. [18]These authors jointly supervised: Brahm H. Segal, Bora E. Baysal. ✉email: Brahm.Segal@RoswellPark.org; Bora.Baysal@RoswellPark.org

  **1**

Macrophages are tissue-localized myeloid cells that originate during embryonic development or from recruited circulating monocytes. Macrophages are known to be active secretory cells that regulate host defense, inflammation, and homeostasis. They can function as antigen-presenting cells that are involved in initiating specific T cell responses[1]. Through phagocytosis, macrophages contain bacterial and fungal infection, clear debris and apoptotic cells, and modulate anti-viral and anti-tumor immunity[2]. Macrophages have a wide range of diversity in their phenotype and functions in response to environmental cues, such as microbial products, products of cellular injury, cytokines, and hypoxia[3]. The biological functions of macrophages are mediated by specific subpopulations that are polarized phenotypically by exposure to specific mediators. Interferon-γ (IFN-γ), TNF-α, or pathogen-associated molecular patterns (PAMPs) can activate pro-inflammatory M1 or classical polarization. M1 macrophages have potent microbicidal activity and release pro-inflammatory cytokines such as IL12 and TNF-α[4,5]. M1 macrophages augment T cell immunity required for control of various intracellular infections (e.g., mycobacteria and Listeria), and can enhance anti-tumor immunity. Alternatively activated (M2) macrophages are involved in anti-inflammatory and immunosuppressive activity through the release of IL10 and synthesis of mediators (e.g., VEGF and Arginase-1) that promote angiogenesis, tissue remolding and wound repair. M2 macrophages can be induced by fungal infections, various parasites, allergy, interleukin-4 (IL4), IL10, IL13 and tumor growth factor beta (TGF-β)[6–8]. While the M1/M2 classification does not capture the plasticity of macrophage phenotypes, it is a widely used designation to study mechanisms for macrophage polarization.

Mitochondrial function, reactive oxygen species (ROS), and metabolic changes modulate macrophage polarization. In general, M1 macrophages depend on glycolysis for ATP production, produce more ROS and accumulate succinate compared to resting macrophage[9,10]. M1 polarization is generally characterized by induction of aerobic glycolysis (Warburg effect) along with high glucose uptake and pyruvate-to-lactate conversion rate. On the other hand, M2 macrophages use oxidative phosphorylation as the predominant source of energy with a lower rate of glycolysis compared to M1 macrophages. Furthermore, forced oxidative phosphorylation in M1 polarized macrophages can skew them to M2 polarization[11–13].

Macrophage responses to environmental cues (e.g., LPS, hypoxia, cytokines) are modulated by complex signaling pathways that include transcriptional regulation[14]. microRNA (miRNAs), DNA methylation and histone modification influence macrophage polarization[15]. In addition to these well-characterized pathways for epigenetic modification, enzyme-regulated RNA editing may be another mechanism regulating macrophage responses. RNA editing is a posttranscriptional mechanism that alters transcript sequences, without affecting the encoding DNA sequences and increases protein sequence diversity[16]. By changing RNA sequences, RNA editing can generate protein diversity in single-cell organisms, plants, and animals[16]. Two main types of RNA editing have been recognized in mammals. One type involves adenine conversion to inosine (A>I) by adenosine deaminase enzymes and occurs in hundreds of thousands of sites located mostly in non-coding, intronic regions[17]. The other dominant type of RNA editing is C>U RNA editing which is catalyzed by the apolipoprotein B-editing catalytic polypeptide-like (APOBEC) family of cytidine deaminase enzymes[18,19].

The initially described activity of APOBEC-mediated C>U editing involved site-specific mRNA editing of ApoB, a protein required for the assembly of very low-density lipoproteins from lipids, by APOBEC1[20]. APOBEC3 proteins have a well-recognized role in inhibiting retroviruses and other viruses in in vitro systems and in restricting the intracellular transposition of retro-elements. This inhibition may be dependent or independent of the deaminase functions of APOBEC3s[21]. Recently, we identified the cellular RNA editing functions of APOBEC3A (A3A)[22] and APOBEC3G (A3G)[23]. A3A and A3G each target specific cytidines located in putative stem-loop structures in distinct sets of transcripts[24]. RNA editing by A3A and A3G can be induced by exogenous over-expression in cell lines[25–27]. A3A-mediated RNA editing is endogenously induced by IFN-γ during M1 macrophage polarization and by type 1 interferons (IFN-1) in monocytes/macrophages[22]. In addition to IFNs, hypoxia and cellular crowding induces endogenous C>U RNA editing by A3A in monocytes and by A3G in natural killer cells[22,23,28]. Hypoxia and IFN-1 additively increase A3A-mediated RNA editing in monocytes, leading to RNA editing levels of ~80% in transcripts of certain genes. A3A-induced or A3G-induced RNA editing by cellular crowding and hypoxia can be mimicked by the inhibition of mitochondrial respiration[23,29]. A3G-mediated RNA editing triggers a Warburg-like metabolic remodeling in HuT78 T cell lymphoma line[23]. These findings suggest that APOBEC3-mediated RNA editing may play a role in hypoxia-induced or IFN-induced cell stress response. Humans have 10 APOBEC genes, but only A3A mediates RNA editing in monocytes and macrophages[22].

In this study, we hypothesized that A3A plays an essential role in macrophage functions during M1 polarization and in response to viral infections. Using primary human monocyte-derived macrophages, we observed A3A-mediated RNA editing during viral infections and M1 polarization with LPS and IFN-γ, and had broad effects on transcriptome, pro-inflammatory and metabolic responses that drive M1 polarization. These results demonstrate a key role for A3A-mediated RNA editing in modulating human macrophage function under ex vivo conditions modeling sepsis and viral infection, and set the foundation for investigating the role of A3A in patients during sepsis and other diseases associated with pathologic macrophage responses.

## Results

**A3A induces *SDHB* RNA editing in M1 macrophages**. We previously showed that IFN-γ and LPS induce RNA editing in M1 macrophages and that the level of editing in several sites is reduced with knockdown (KD) of A3A[22]. Here, M0 cells were treated for 2 days with 20 ng/ml recombinant human IFN-γ and 50 ng/ml *Escherichia coli* LPS, or 20 ng/ml recombinant human IL4, for M1 or M2 polarization, respectively. In KD experiments, M0 macrophages were transfected with negative control scrambled (SC) or A3A siRNAs 1 day before induction of M1 polarization. To examine the induction of A3A-mediated RNA editing, we used the previously developed a RT-qPCR assay for the c.C136U (R46X) event in succinate dehydrogenase (mitochondrial complex II) subunit *SDHB* mRNA[28]. We first confirmed that the induction of *SDHB* c.C136U RNA editing occurs in M1 macrophages but not in M0 or M2 macrophages (Fig. 1a). To evaluate whether A3A mediates *SDHB* c.C136U RNA editing in M1 macrophages, we knocked down A3A by siRNA and induced the polarization of M0 macrophages to M1 macrophages with LPS and IFN-γ. KD of A3A led to a statistically significant reduction in the level of *SDHB* c.C136U RNA editing as tested in eight additional donor samples (Fig. 1b). These findings demonstrate that A3A is responsible for the observed *SDHB* c.C136U RNA editing in M1 macrophages. Macrophage viability, assessed by flow cytometry, was similar in M1 A3A KD and M1 A3A SC macrophages (Supplementary Fig. 1).

**IFN-1 induces RNA editing in tumor-associated macrophages (TAMs)**. After showing the induction of A3A-induced RNA editing in normal macrophages in the context of M1 polarization, we asked whether A3A modulates responses in TAMs. Although TAMs are generally M2-like, the tumor microenvironment is often hypoxic and characterized by pro-inflammatory cytokines, including

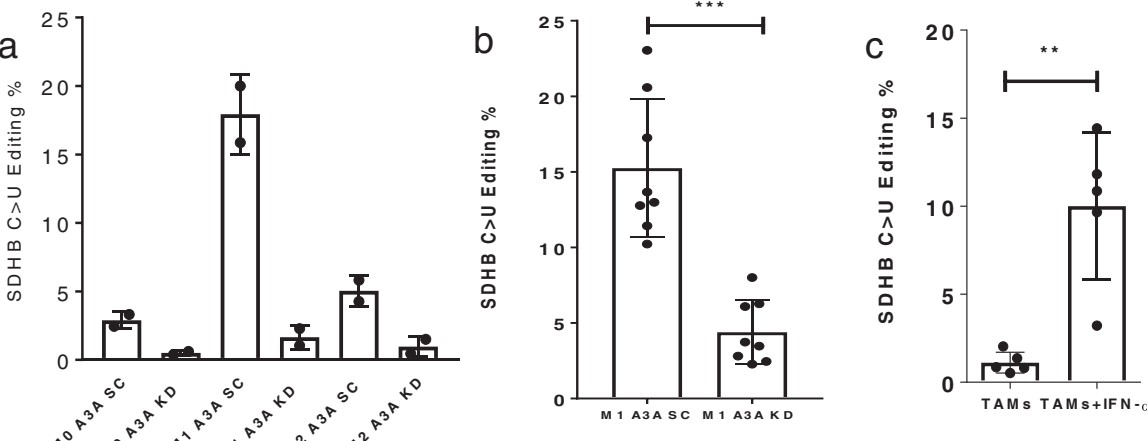

**Fig. 1 _SDHB_ c.C136U RNA editing is induced in M1 macrophages and mediated by A3A. a** RNA editing was induced in M1 macrophages but not in M0 or M2 macrophages ($n = 2$ replicates from 1 representative donor). **b** A3A was required for _SDHB_ RNA editing. $P = 0.0001$, Paired $t$ test, two-tailed, $n = 8$ donors. **c** IFN-1 (IFN-$\alpha$, 1000 U/ml) induces RNA editing in ovarian tumor associated macrophages (TAMs) from ascites of ovarian cancer patients. $P = 0.0084$, Paired $t$ test, two-tailed, $n = 5$ donors. Data in scatter dot plots with mean ± SD. (see "Methods" section for macrophage polarization and KD protocol.).

interferon. We investigated the impact of IFN-1 on TAMs purified from the ascites of five patients with newly diagnosed epithelial ovarian cancer. TAMs were isolated by CD14 microbeads, and purity was confirmed by CD33[hi] expression and by cytology. Compared to untreated samples, we found that editing level was induced significantly in TAMs treated with IFN-1 (Fig. 1c). These results show that A3A-mediated RNA editing is also inducible in TAMs by IFN-1.

**Viral infections induce RNA editing by A3A in primary human M0 macrophages.** Interferons are induced by viral infections and are required for antiviral defense. We therefore tested whether viral infection would induce A3A-mediated RNA editing in M0 macrophages. Normal donor monocyte-derived macrophages were generated as described above and infected with two clinically relevant viruses: Maraba and influenza. Maraba is a genetically modified SS-RNA oncolytic rhabdovirus being developed for cancer therapy[30,31]. Influenza is a SS-RNA orthomyxovirus. We observed that viral infection induced _SDHB_ RNA editing (Fig. 2a). Then we tested whether induction of RNA editing in M0-infected was A3A-dependent. Normal donor monocyte-derived macrophages were generated as described above and A3A was knocked down 24 h before infection with Maraba or influenza virus. We observed that RNA editing level was suppressed in M0 A3A KD compared to M0 A3A SC (Fig. 2a). A higher concentration of IFN-$\alpha$ was detected in the supernatant of M0 macrophages infected with Maraba and influenza viruses but not in non-infected M0 macrophages (Fig. 2b). A3A and KD siRNA transfection did not induce IFN-$\alpha$ secretion, consistent with a lack of _SDHB_ RNA editing in M0 A3A SC and M0 A3A KD macrophages (Figs. 1a and 2a). M0 macrophage viability, assessed by trypan blue, reduced after influenza virus infection (Supplementary Fig. 2). These results show that viruses can induce A3A-mediated RNA editing in macrophages, likely mediated by increased IFN production that is known to induce A3A expression levels by hundreds-fold[22].

**APOBEC3A catalyzes the majority of C>U RNA editing events during M1 polarization.** We next tested the extent that A3A mediated C>U RNA editing during M1 polarization. Monocyte-derived macrophages from three normal donors were transfected with A3A KD and A3A control SC siRNAs followed by induction

of M1 polarization. Non-polarized macrophages (M0) were used as specificity controls. RNA editing of _SDHB_ c.C136U was confirmed by RT-qPCR and ranged between 11% and 21% in M1 A3A SC- versus 1–3% in M1 A3A KD macrophages (RNA editing in M0 macrophages was 0–2%). Paired RNA_Seq analysis was performed comparing M1 A3A SC and M1 A3A KD macrophages from the same donor.

We first determined the distribution of RNA mismatches upregulated >2-fold in SC M1 vs. M0 and A3A KD M1 vs. M0, and found that KD of A3A caused greater decreases in C>T/G>A events than the other types of RNA mismatches (Supplementary Fig. 3). To directly test the extent that A3A catalyzes these RNA editing events, we first determined the C>U RNA editing events that were induced at least 2-fold in M1 SC siRNA relative to the control M0 samples and that occur at >5% level in any experimental group, using high-coverage RNA_Seq data and stringent filtering steps described in the "Methods" section. This analysis revealed 209 C>U RNA-editing sites in 203 genes in SC M1-macrophages (Supplementary Data 1). KD of A3A during M1 polarization reduced RNA editing levels by more than 2-fold in 180 of these 209 (~86%) sites, indicating that A3A catalyzes the majority of C>U RNA editing sites during M1 polarization (Fig. 3a). The RNA editing sites that were not affected by A3A KD may represent genomic SNVs, true RNA-edited sites that are not catalyzed by A3A, or false positives. About half of the C>U RNA-editing events catalyzed by A3A were synonymous ($98/180 = 54.4\%$), followed by non-synonymous ($36/180 = 20\%$) and 3'-UTR ($33/180 = 18.3\%$) (Fig. 3b, c).

Examination of the non-synonymous/stop-gain RNA editing events (Supplementary Data 1) showed the highest editing level in _THOC5_ with 75% RNA editing level in M1 A3A SC macrophages. The mean RNA editing level of _THOC5_ was reduced to about 32% in M1 A3A KD macrophages versus A3A SC (Table 1). _THOC5_ RNA editing (c.C2047T:p.R683C) alters the last amino acid arginine, which is conserved in all sequences from 100 vertebrate genomes (Supplementary Fig. 4a). In addition, the edited C and the surrounding nucleotides that predict a stem-loop RNA structure are also highly conserved, including the −1 position, which is either T or C (Supplementary Fig. 4b). Bioinformatic analyses by PROVEAN, SIFT and POLYPHEN-2 methods, respectively, predict "deleterious", "damaging" and "probably damaging" effects of the R683C protein variation. Western blot analysis of M1 SC control and M1

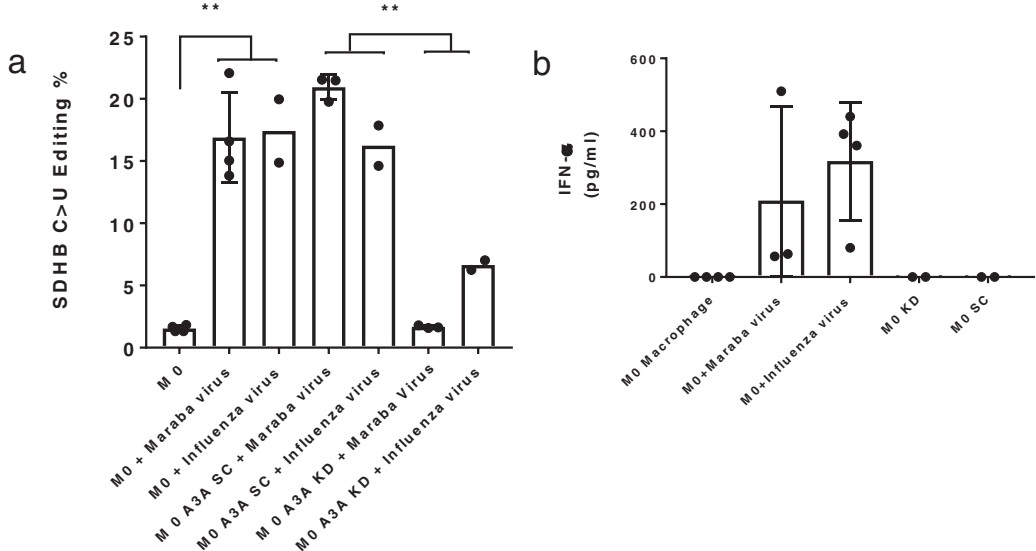

**Fig. 2 Viruses induce A3A-mediated RNA editing in macrophages.** Circulating healthy donor monocytes (~1 × 10⁶ cells per well) were differentiated into M0 macrophages followed by siRNA transfection (A3A knock-down [KD] or scramble [SC]). After 24 h, were infected with Maraba (10⁶ PFU per well) or influenza virus (2 × 10⁴ PFU per well, MOI = 0.02), and *SDHB* RNA editing was assessed at 48 h after infection: $n = 3$ donors for Maraba and $n = 2$ donors for influenza viruses. **a** Viral infection of M0 macrophages increased *SDHB* RNA editing levels. **$P = 0.0095$, Mann–Whitney test, two-tailed. A3A KD significantly reduced *SDHB* RNA editing levels induced by viral infections. **$P = 0.0079$, Mann–Whitney test, two-tailed. **b** IFN-α levels in culture supernatant increased after Maraba or influenza virus infections, but were undetectable after siRNA (SC or A3A KD) transfections. $n = 2$–4 donors. Data in scatter dot plots with mean ± SD.

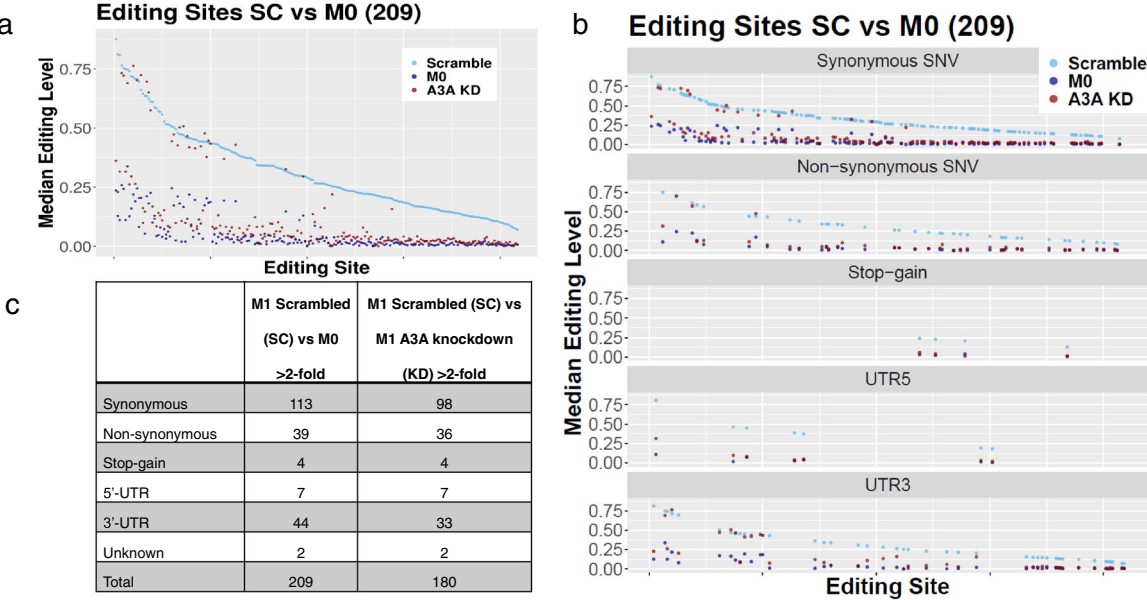

**Fig. 3 A3A catalyzes the majority of C>U RNA editing events during M1 polarization. a** A3A KD reduces the C>U RNA editing levels in SC M1 macrophages close to levels seen in M0 unpolarized macrophages. C>U RNA editing sites are ranked by their levels in M1 A3A SC control. **b** and **c** C>U RNA editing events in M1 A3A SC, M1 A3A KD and M0 cells are separately shown for variant types in graph.

A3A KD samples shows reduced expression of A3A protein by siRNA KD, but similar amounts of THOC5 protein, suggesting that missense alteration by RNA editing does not alter THOC5 protein stability (Fig. 4a). Other genes including *ZNF124, ARSB, ATXN2, SLC37A2, GAA, PCGF3, CDYL2,* and *S1PR2* also showed high levels (>20%) of non-synonymous RNA editing in M1 A3A SC compared to M1 A3A KD macrophages (Table 1).

Sanger sequencing of RT-PCR products obtained from a different monocyte donor confirmed high-level (~50%) of RNA editing in *THOC5* and *ZNF124* gene transcripts during M1 polarization and reduced editing levels by A3A KD. As expected, M0 macrophages showed no evidence of RNA editing (Fig. 4b). Together, these results demonstrate that mRNAs of certain genes are substantially altered by A3A-mediated RNA editing in M1-polarized macrophages.

**A3A upregulates pro-inflammatory gene expression in M1 macrophages.** Relative to A3A SC, A3A KD caused more than 2-fold downregulation of 81 genes and upregulation of 131 genes

**Table 1 Gene transcripts that acquire high levels of non-synonymous C>U RNA editing in M1 macrophages.**

| Gene | Genomic coordinate of RNA editing (GRCh38/hg38) | RNA editing event | SC average editing level | A3A KD average editing level | SC/KD ratio |
|---|---|---|---|---|---|
| SDHB | chr1:17044825 | NM_003000:exon2:c.C136T:p.R46X | 0.227 | 0.040 | 5.70 |
| THOC5 | chr22:29508462 | NM_001002879:exon20:c.C2047T:p.R683C | 0.750 | 0.316 | 2.38 |
| ZNF124 | chr1:247157042 | NM_001297568:exon4:c.C580T:p.R194C | 0.589 | 0.117 | 5.04 |
| ARSB | chr5:78984968 | NM_198709:exon2:c.C281T:p.S94L | 0.434 | 0.074 | 5.86 |
| ATXN2 | chr12:111516348 | NM_001310121:exon10:c.C866T:p.S289L | 0.396 | 0.049 | 8.04 |
| SLC37A2 | chr11:125080652 | NM_198277:exon7:c.C566T:p.S189F | 0.342 | 0.049 | 7.00 |
| GAA | chr17:80107625 | NM_001079804:exon4:c.C761T:p.S254L | 0.341 | 0.021 | 16.39 |
| PCGF3 | chr4:743578 | NM_006315:exon7:c.C367T:p.R123W | 0.300 | 0.066 | 4.55 |
| CDYL2 | chr16:80684900 | NM_152342:exon2:c.C254T:p.S85L | 0.248 | 0.018 | 13.77 |
| S1PR2 | chr19:10224709 | NM_004230:exon2:c.C197T:p.S66L | 0.220 | 0 | N/A |

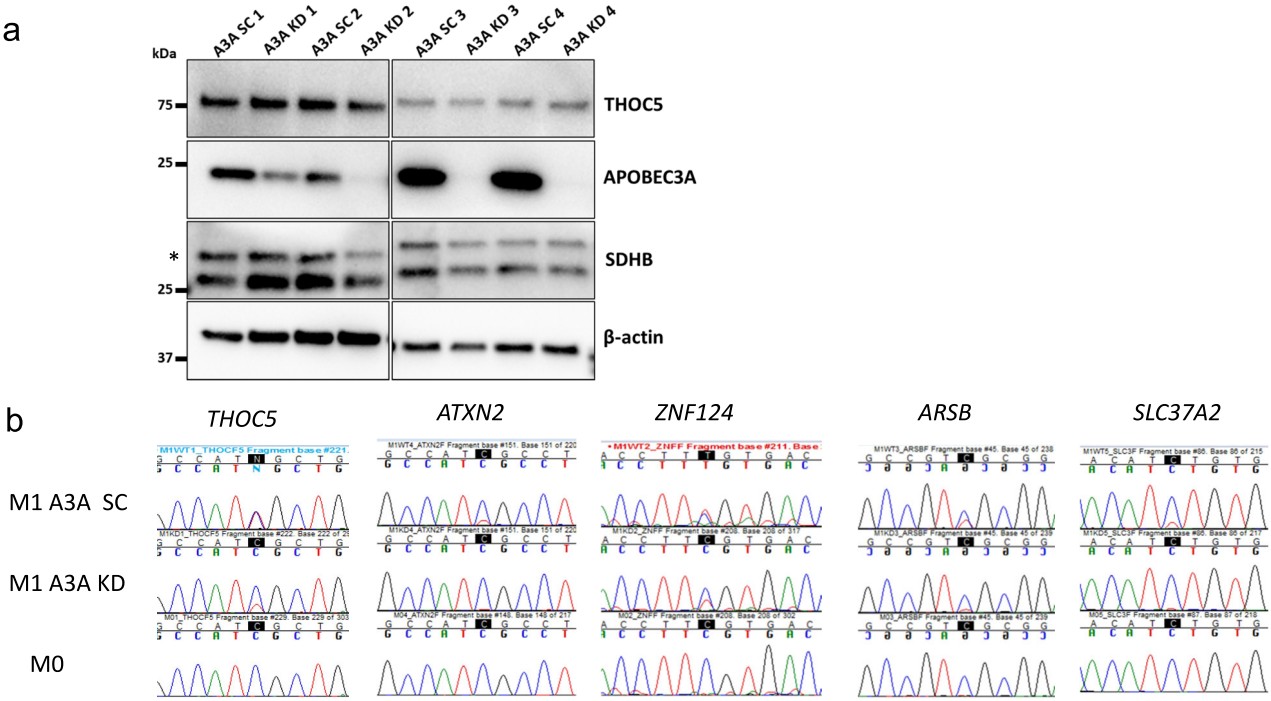

**Fig. 4 Knock down of APOBEC3A reduces APOBEC3A protein expression and C>U RNA editing levels in M1 macrophages. a** THOC5, APOBEC3A and SDHB (upper band with asterisk) protein levels in whole-cell lysates (20 μg protein) of M1 macrophages isolated from A3A SC and A3A KD samples from four separate set of donors are shown. Lanes for APOBEC3A and SDHB were run in parallel in the same gel. **b** Sanger sequencing of selected gene mRNAs confirms the induction of RNA editing in M1 A3A SC, which is reduced by A3A KD. Unpolarized M0 samples show no evidence for RNA editing. The edited Cs are highlighted within black boxes.

with statistical significance (Padj < 0.05; Supplementary Data 2). A3A was the second most downregulated gene upon A3A KD, confirming the intended siRNA targeting. Pro-inflammatory cytokines *IL6*, *IL23A*, and *IL2B* were among the most down-regulated genes upon A3A KD in M1 macrophages (Fig. 5 and Supplementary Fig. 5a). Examination of selected inflammation-related genes showed differences between M1 A3A SC and M1 A3A KD. A3A KD reduced the expression of cell surface pro-inflammatory genes *CD68*, *CD80* and *CD86* but increased the expression of *MRC1* (*CD206*), a marker of M2 polarization. The expression of inflammatory cytokine genes *TNF*, *IL6*, *IL23A*, *IL18*, and *CXCL8* was also statistically significantly reduced by A3A KD (Supplementary Fig. 5b). Gene ontology (GO) analysis showed that A3A augmented the expression of pathways driving cytokine/chemokine production and signaling, adaptive immunity, and cell death, but markedly down-regulated pathways

involved in protein elongation/translation and metabolism and influenza viral RNA transcription and replication (Table 2). These results suggest that A3A plays an important role in the regulation of gene expression in M1 polarized macrophages, with the major effects being upregulation of pro-inflammatory genes and pathways while reducing expression of genes regulating protein synthesis, metabolism, and influenza life cycle.

**A3A enhances the release of inflammatory cytokines and CD86 expression in M1 macrophages.** To examine whether the pro-inflammatory role of A3A observed at the gene expression level also extends into protein expression level, we evaluated the role of A3A in macrophage surface antigen expression and cytokine responses. KD of A3A was confirmed by reduction in *SDHB* c. C136U RNA-editing levels (Fig. 1). Cytokine production by M1

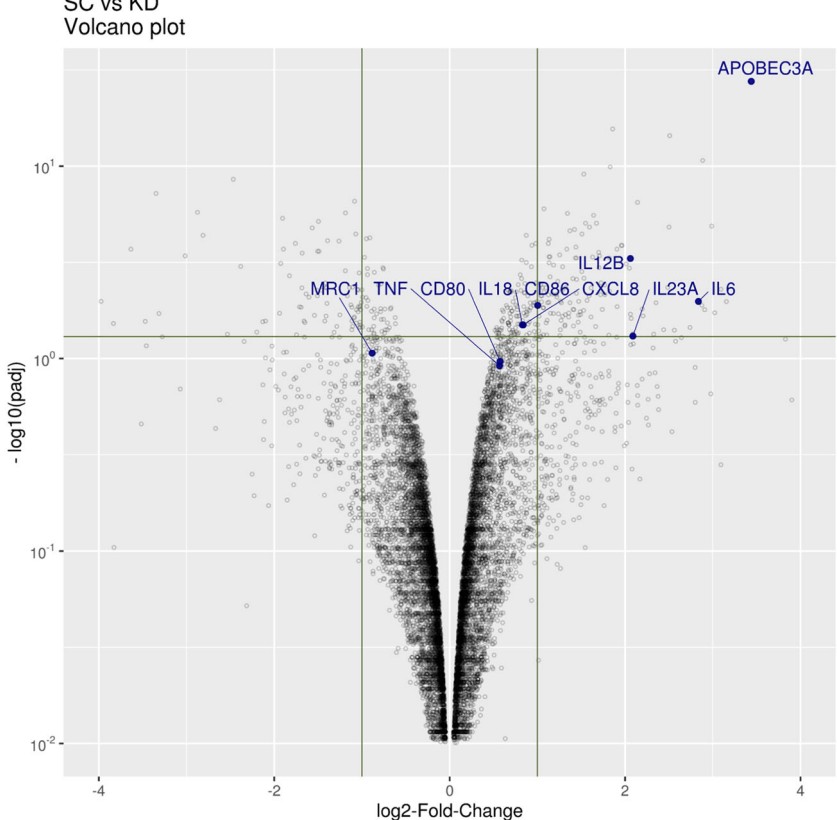

**Fig. 5 A3A augments inflammatory gene expression in M1-polarized macrophages.** Volcano plot of SC vs. A3A KD macrophages highlight selected inflammatory genes. Positive log2FC values represent enriched expression in SC and negative values enriched in KD. Plot highlights that A3A expression is significantly enriched in SC relative to KD, indicating effective knockdown and that A3A is required for high expression levels of pro-inflammatory genes *IL6*, *IL23A*, *IL12B* and *CD86* in M1 macrophages. Vertical lines indicate absolute 2-fold-change values and the horizontal line FDR = 0.05.

**Table 2 Pathways upregulated and downregulated by A3A in M1-polarized macrophages.**

| Upregulated | Downregulated |
|---|---|
| Adaptive immune system**** | Peptide elongation/translation**** |
| Cytokine–cytokine receptor**** | Glycolysis-gluconeogenesis* |
| G-protein-coupled receptor**** | Pyruvate metabolism* |
| Interferon alpha-beta**** | PID-HIF1-TF pathway* |
| Antigen processing and presentation**** | Mitochondrial protein** |
| Class-1-MHC mediated**** | Protein metabolism**** |
| Apoptosis** | Pentose phosphate pathway* |
| Chemokines receptors**** | Arginine and proline metabolism* |
| Stress pathway* | Influenza life cycle**** |
| Cytokines pathway** | Influenza viral RNA transcription and replication**** |
| MHC-Class-II antigen presentation*** | Integrin-1 pathway** |
| Death pathway* | Transport of glucose* |

FDR q-val by GO analysis comparing A3A SC vs. A3A KD: * <0.05; ** <0.01; *** <0.001; **** <0.001.

A3A SC and M1 A3A KD macrophages was assessed by ELISA. A3A KD decreased TNF-α, IL-1β, and IL6 secretion by M1 macrophages (Fig. 6a–c).

We next assessed the role of A3A in modulating the surface expression of CD86 and CD206 as markers for M1 macrophages and M2 macrophages, respectively, by flow cytometry[32]. Compared to M1 A3A SC macrophages, M1 A3A KD macrophages showed a reduction in CD86 expression (Fig. 6d). CD206 surface expression did not change in M1 A3A KD compared to M1 A3A SC macrophages (Supplementary Fig. 6). As expected, control M2-polarized macrophages showed reduced CD86 and increased CD206 expression. These results demonstrated that A3A augments pro-inflammatory responses in M1 macrophages.

**A3A suppresses glycolysis in M1 macrophages.** While resting macrophages rely principally on mitochondrial respiration for energy, the switch to glycolysis under aerobic conditions is associated with polarization to M1 macrophages[32]. Since A3A plays a role in the augmentation of inflammatory responses of M1 macrophages, we hypothesized that A3A induces glycolysis during M1 polarization. We evaluated glycolysis and mitochondrial respiration in M1 A3A SC and M1 A3A KD macrophages by the SeaHorse Glycolytic Stress and Mitochondrial Stress assays, respectively. As expected, M1 macrophages had a higher glycolysis rate compared to M2 macrophages. However, the level of glycolysis was increased in A3A KD M1 macrophages compared to M1 A3A SC, indicating that A3A reduced glycolysis in M1 macrophages (Fig. 7a). These results are consistent with our GO analysis (Table 1) showing that A3A reduces expression of genes involved in glycolysis and other metabolic pathways. Basal oxygen consumption rates were similar between M1 A3A KD and M1 A3A SC macrophages (Fig. 7b). Since mitochondrial complex II participates in oxidative phosphorylation, we also tested SDHB protein expression. *SDHB* mRNA acquires 10–15% nonsense c. C136U (R46X) RNA editing. We found no differences in *SDHB*

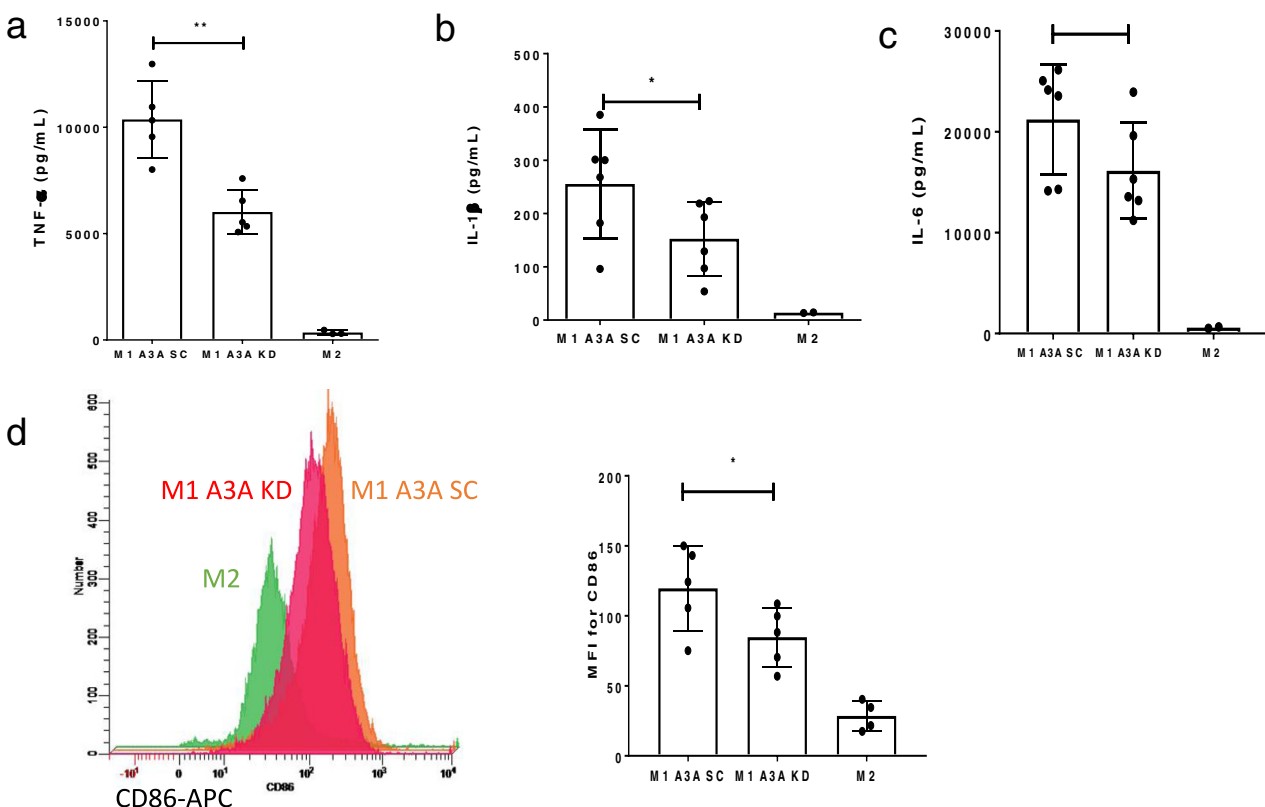

**Fig. 6 Knockdown of A3A impairs pro-inflammatory phenotype of M1 macrophages. a** A3A KD decreases TNF-α secretion by M1 macrophages from normal donors compared to controls (SC). $P = 0.0049$, Paired $t$ test, two-tailed, $n = 5$ donors. **b** A3A KD decreases IL-1β secretion by M1 macrophages from normal donors compared to controls (SC). $P = 0.0103$, Paired $t$ test, two-tailed, $n = 6$ donors. **c** A3A KD decreases IL6 secretion by M1 macrophages from normal donors compared to controls (SC). $P = 0.0192$, Paired $t$ test, two-tailed, $n = 6$ donors. **d** A3A KD reduces CD86 surface protein expression in M1-macrophages by flow cytometry. Cytokines are measured by ELISA (see the "Methods" section). **e** The quantification of CD86 ($P = 0.0395$, Paired $t$ test, two-tailed, $n = 5$ donors) is made using flow cytometry (see the "Methods" section) following M1 polarization in CD33-positive cells. MFI = mean (geometric) Fluorescent Intensity. M2 macrophage data from a smaller number of donors ($n = 2$-4) highlight low expression levels of inflammatory markers in M2. Data in scatter dot plots with mean ± SD.

protein expression levels between M1 A3A KD and M1 A3A SC macrophages (Fig. 4a). These results suggest that A3A enhances the production of pro-inflammatory cytokines independently of the Warburg effect and basal $O_2$ consumption rates in M1 macrophages.

## Discussion

We provide evidence that A3A is required for transcriptomic and functional polarization of M1 macrophages, most likely through C>U RNA editing of scores of genes. We show that site-specific C>U RNA editing by A3A can be induced by M1 polarization or viral infections in normal monocyte-derived macrophages, and by IFN-1 exposure in TAMs isolated from ovarian cancer-related ascites fluid. Together, these results point to a broad role for A3A in modulating macrophage responses to diverse conditions.

We observed that viruses also induce A3A-dependent cellular RNA editing, which, in turn, downregulates cellular transcripts that support viral (influenza) replication (Table 2). IFN-1 plays a role in M1 polarization of macrophages exposed to virus infection[33]. Accordingly, IFN-1 concentration was detectable in the media of macrophages infected by the Maraba and influenza viruses (Fig. 2b). MG1 Maraba is an oncolytic virus that is being tested in phase1/2 clinical trials, and acts through multiple anti-tumor mechanisms including direct tumor cell lysis, modulation of tumor microenvironment to a more pro-inflammatory state

and generation of tumor-specific T-cells[34,35]. Our findings raise the speculation that when TAMs are infected by Maraba virus, A3A promotes their pro-inflammatory polarization, creating a favorable microenvironment for development of antitumor immunity, while at the same time enhancing antiviral response. Therefore, the role of A3A-mediated RNA editing in oncolytic virus therapy for cancer requires further studies.

RNA-Seq analysis showed that A3A increases gene expression of pro-inflammatory cytokines like *TNF, IL6, IL18*, and *CXCL8*, while also limiting the glycolysis in M1 macrophages. GO analysis showed that A3A down-regulated the glycolysis pathway and metabolic pathways involved in protein translation, glucose, amino acid, and lipid metabolism, and HIF-1 in M1 macrophages. These results demonstrate that A3A enhances the pro-inflammatory responses in M1 macrophages, likely at the expense of genes mediating metabolic responses. Given that A3A is located in the cytoplasm without translocating to the nucleus in normal monocytic cells[36], and no evidence yet exists that A3A can selectively deaminate nuclear DNA in normal cells, these findings strongly suggest that A3A's RNA editing function is responsible for metabolic, gene expression, cytokine secretion and surface marker expression changes associated with M1 polarization. Since A3A-mediated RNA editing is also induced by cellular crowding and hypoxia in normal monocytes[22], we speculate that C>U RNA editing plays a general role in mono-cyte/macrophage stress response under a variety of physiologically relevant conditions.

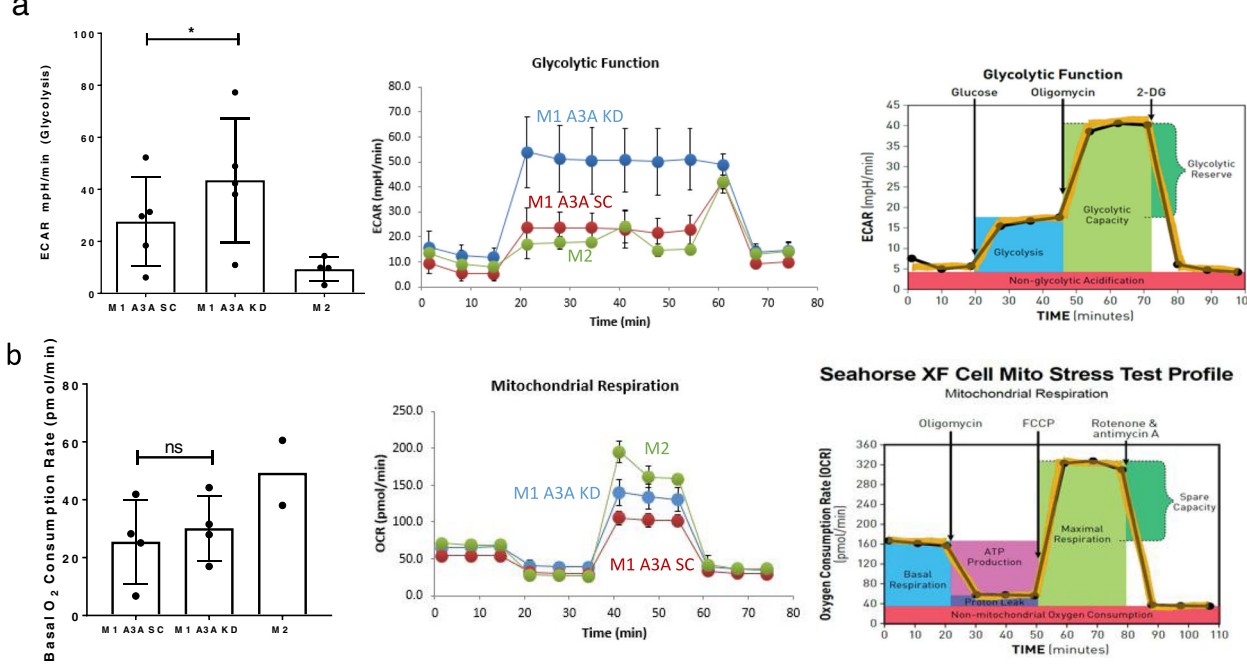

**Fig. 7 The role of A3A in glycolysis and oxygen consumption in macrophages. a** A3A KD enhances glycolysis (ECAR) compared to A3A SC M1 macrophages derived from normal donor blood. *P* = 0.0171, Paired *t* test of glycolysis values, two-tailed, five donors. A representative original trace from SeaHorse platform shows ECAR in M1 A3A KD cells relative to control M1 A3A SC. **b** Basal oxygen consumption rate is similar between M1 A3A KD and M1 A3A SC macrophages. *P* not significant (*P* = 0.1358), Paired *t* test of basal oxygen consumption values, two-tailed, four donors. A representative original trace from SeaHorse platform shows basal oxygen consumption in M1 A3A KD cells relative to control M1 A3A SC. M2 macrophage data from a smaller number of donors (*n* = 2, 4) are shown for comparison. Data in scatter dot plots with mean ± SD.

Cell stress is commonly encountered by innate immune cells in pathologic microenvironments of infection and inflammation. Sudden environmental changes can induce cell stress, which triggers pathways for homeostasis or cell death, depending on the degree of cellular damage. Cells have adaptive stress responses including DNA repair, metabolic reprogramming, and UPR pathways that enable survival during states of emergency, such as infections and hypoxia[37]. Constitutive activation of IFN-1 signaling can cause tissue dysfunction and organ failure in certain diseases, such as Aicardi–Goutières Syndrome[38]. Our results support distinct roles for A3A in the adaptation of macrophages to specific stressors. In response to stressors LPS+IFN-γ, IFN-1 or viruses, A3A augments M1 polarization and pro-inflammatory responses (Figs. 2, 5 and 6). Our results raise the notion that while in some settings A3A may augment macrophage-mediated host defense through amplification of inflammation, A3A may be deleterious in other conditions associated with inflammatory injury, such as sepsis. While a shift from oxidative phosphorylation to aerobic glycolysis occurs in leukocytes during the initial host response to sepsis, a generalized metabolic defect involving both glycolysis and oxidative metabolism characterizes later stages of sepsis[39]. Seen in this light, our results point to A3A as a potential contributor to defects in energy metabolism associated with sepsis.

We note some functional parallels between AID (activation-induced cytidine deaminase encoded by AICDA) and APOBEC3A in immunity. AID and APOBEC3A are evolutionarily related cytidine deaminases[40] that physiologically alter the genomically encoded cellular DNA or RNA sequences as part of adaptive and innate immunity responses, respectively. AID is specifically expressed in germinal center B cells and mediates somatic hypermutation of Ig DNA, leading to generation of higher affinity antibodies in terminally differentiated memory B cells or plasma cells[41]. In contrast, APOBEC3A is primarily expressed in monocyte/macrophages and its inducible RNA editing function promotes M1 pro-inflammatory phenotype in pathologic microenvironments.

A3A-mediated DNA mutations were previously identified in human cancer cells[42,43]. We previously examined genomic DNAs corresponding to 23C>U RNA editing sites in primary human monocytes treated with hypoxia/IFN1 (*n* = 2 donors) and in 293T/APOBEC3A overexpression system (*n* = 2 replicates)[22]. Separately, we cloned PCR-amplified exon 2 of SDHB genomic DNA, which contains the RNA editing site, in 293T/APOBEC3A overexpression system and sequenced individual clones[25]. In both of these studies, we found no evidence of DNA mutations at sites corresponding to APOBEC3A-induced RNA-editing sites. Therefore, we consider cellular RNA editing as the principle mechanism for A3A modulating macrophage function.

The mechanism by which cellular RNA editing by A3A promotes M1 polarization requires further investigation. C>U RNA editing by A3A may alter certain gene products or affect the stability or translatability of mRNAs, which in turn causes a cascade of downstream effects to facilitate the M1 phenotype polarization. Using high-coverage RNA_Seq data, we find a greater number of exonic/UTR C>U RNA editing sites (*n* = 209) than initially suggested by analysis of publicly available RNA_Seq data (*n* = 120) in M1 macrophages[22], including the identification of THOC5 as the most nonsynonymously edited gene. THOC5 encodes a member of the THO complex that is part of the mRNA transcription/export complex[44]. THOC5 protein is phosphorylated by several tyrosine kinases such as the M-CSF receptor[45]. Tran et al. [46] showed that depletion of THOC5 in mouse bone marrow-derived macrophages downregulated 99 genes and suppressed M-CSF-mediated M2-like macrophage differentiation by inhibiting the export of M-CSF inducible genes. It is therefore conceivable that one mechanism by which A3A-mediated RNA` editing promotes M1 polarization is through physiologic suppression of THOC5, which leads to reduced

M-CSF-regulated gene export. Our study opens new avenues of investigation of the molecular basis by which A3A regulates inflammation and metabolism in macrophages and the potential of A3A as a therapeutic target for diseases associated with pathologic inflammation.

## Methods

**Macrophage generation and polarization.** Normal human donor CD14+ monocytes were isolated from TRIMA leukoreduction filters following platelet apheresis at the blood donor center at Roswell Park Comprehensive Cancer Center (RPCCC). Since these cells were remnant products of platelet apheresis and contained no identifiers, their use was considered for non-human research. Mononuclear cells were recovered with lymphocyte separation media and sepMate tubes (Stemcell Technology, Vancouver, Canada) by centrifugation. Monocytes were isolated from mononuclear cells using CD14 microbeads and AutoMACS (Miltenyi Biotec, Somerville, MA). The purity of monocytes was confirmed by cytology and flow cytometry (>90% based on CD33+ CD14+ expression). Cells were cultured for 1 week at a density of $10^6$/well in six-well plates with 50 ng/ml recombinant human macrophage colony-stimulating factor (M-CSF) (Life Technologies, Carlsbad, CA), 1× GlutaMAX-I (Life Technologies), and 1 mM sodium pyruvate (Mediatech, Manassas, VA) to generate M0 macrophages. For M1 or M2 macrophage polarization, M0 cells were treated for 2 days with 20 ng/ml recombinant human IFN-γ (Life Technologies) and 50 ng/ml *Escherichia coli* LPS (Sigma Aldrich, St. Louis, MO), or 20 ng/ml recombinant human IL4 (Life Technologies), respectively. RNA was isolated from cells using TRIzol™ Reagent (Invitrogen, Carlsbad, CA)[22]. Research protocol for tissue collection and subsequent work for RNA editing was approved by RPCCC's IRB.

**KD of A3A in macrophages.** A day before induction of M1 polarization, M0 macrophages were transfected with 100 nM of negative control (Silencer negative control no. 1, product number AM4611, Life Technologies) or equimolar mix of two human APOBEC3A siRNAs (Silencer 45715 and 45810, respectively, with sense sequences 5′-GACCUACCUGUGCUACGAATT-3′ and 5′-GCAGUAUGC UCCCGAUCAATT-3′, Life Technologies) using Lipofectamine RNAiMAX (Life Technologies), following the manufacturer's protocol. IFN-γ and LPS were added with 2 ml medium to each well, to induce M1 polarization, and cells were harvested at 48 h[22].

**Reverse transcription and PCR.** RNA was reverse-transcribed with random DNA hexamers and oligo-dT primers using material and methods provided with the Transcriptor First Strand cDNA Synthesis (Roche, Indianapolis, IN). PCR typically employed 35 cycles of amplification and an annealing temperature of 60 °C. Primers (Integrated DNA Technologies Coralville, IA) used for PCR of cDNA templates were designed such that the amplicons spanned multiple exons. A blend of Taq and high-fidelity Deep VentR DNA polymerases (OneTaq, New England Biolabs) was used in PCR to generate products for Sanger sequencing. KD of A3A is confirmed by RT-qPCR using allele-specific primers and *SDHB* probe on a Light Cycler 480 System (Roche), as previously described[22,28]. Quantification cycle (Cq) values were calculated by the instrument software using the maximum second derivative method, and the mean Cq value of duplicate or triplicate PCR reactions was used for analysis[22].

**Cell lysis and Western blot.** Cells were washed with 1× phosphate-buffered saline (PBS) and whole-cell lysates were collected using Pierce RIPA buffer (Thermo Fisher, Grand Island, NY, Catalog number: 89900) with 1× Halt protease and phosphatase inhibitor cocktail (Thermo Fisher, Catalog number: 78440). Protein concentrations were quantified using Pierce Coomassie (Bradford) Protein Assay Kit (Thermo Fisher, Catalog number: 23200). 20 µg proteins were mixed in Laemmli buffer, denaturized at 95 °C for 5 min and run on SDS–PAGE using a pre-cast, 4–20% gradient polyacrylamide gels (Mini-PROTEAN TGX, Bio-Rad, Hercules, California). Proteins were then transferred to polyvinylidene difluoride membrane (Immobilon-P PVDF Membrane, EMD Millipore, St. Louis, MO) at a constant voltage of 100 V for 70 min at 4 °C using Mini Trans-Blot® Cell (Bio-Rad). Membranes were incubated in Tris-buffered saline (TBS) with 0.05% v/v TWEEN 20 (Sigma Aldrich) and 5% w/v nonfat dry milk (Blotting-Grade Blocker #1706404, Bio-Rad). Primary antibodies were applied to the PVDF membrane using the indicated dilutions: Rabbit polyclonal Anti-PHO1 (APOBEC3A) antibody (Abcam, Cambridge, MA, product number ab262853, 1:1000 dilution), Rabbit polyclonal anti-THOC5 antibody (Bethyl Laboratories, product number A302-120A, 1:1000 dilution), Mouse monoclonal anti-SDHB antibody (Santa Cruz, Dallas, TX, product number sc-271548, 1:500 dilution), and mouse monoclonal anti-beta actin antibody (Abcam, product number ab49900, 1:1000 dilution). Horseradish peroxidase-conjugated, donkey anti-rabbit (Fisher Scientific, Catalog number: 45-000-682) or goat anti-mouse (Sigma-Aldrich, Catalog number: A4416). IgG antibodies were used at 1:2000 dilution. Pierce ECL western blotting substrate (Thermo Scientific, Catalog number: 32106) was used for chemiluminescent detection. Signals were visualized and imaged using the ChemiDoc XRS+ System and Image Lab Software (Bio-Rad).

### Viral infections

*Human monocyte-derived macrophage influenza exposure.* Monocytes were isolated from freshly drawn blood, suspended in 1× GlutaMAX-I (Life Technologies) and 1 mM sodium pyruvate (Mediatech) medium at $3.3 \times 10^5$ cells/ml, and 3 ml was dispensed into each well of six-well tissue culture plates. M-CSF, 50 ng/ml (Life Technologies), was added and the cultures were incubated at 37 °C, 95% relative humidity, and 5% $CO_2$ for 7 days. Influenza inoculum was prepared by thawing a stock vial of influenza A/Brisbane/10/2007 (H3N2) (a generous gift from Dr. Suryaprakash Sambhara at the Center for Disease Control and Prevention, Atlanta, GA), sonicated for 45 s in an ultrasonic water bath (Branson 1210, VWR, Radnor, PA), and diluted to $2 \times 10^5$ PFU/ml (as determined in MDCK plaque assay) with ice cold PBS. The plated cells were washed 2× with 1 ml warm 3% BSA in DMEM after which 0.3 ml was added to prevent drying of the cells. 100 µl of influenza inoculum (MOI = 0.02) or PBS control was added to appropriate wells and adsorbed for 1 h at 37 °C, 95% relative humidity, and 5% $CO_2$ while gently rocking. Warm medium, 2.6 ml, was added to each well and incubation continued. After 24 or 48 h, the medium was collected, centrifuged at $500 \times g$ for 5 min at 4 °C, and the supernatant was quick frozen in ethanol and dry ice and stored at −80 °C for influenza MDCK plaque assay analysis. The cells remaining in the well were harvested by adding 1 ml of Trizol to each well, incubating at room temperature for 5 min and transferring to a microfuge tube and storing at −80 °C for RNA analysis. In vitro studies with influenza virus was approved by University at Buffalo's Institutional Biosafety Committee.

*Influenza MDCK plaque assay.* MDCK cells (ATTC, Manassas, VA) were seeded on six-well tissue culture plates at $4 \times 10^5$ cells/well in 2 ml of MEM with Earle's salts + 0.1 mM non-essential amino acids + 1 mM pyruvate + 50 U/ml penicillin + 50 µg/ml + streptomycin + 20 µg/ml gentamicin + 10% fetal calf serum and incubated at 37 °C, 95% relative humidity, and 5% $CO_2$. When cells reached 80–90% confluency (≈3 days) the cells were rinsed 2× with 1 ml 0.3% BSA in DMEM and 0.3 ml was added to prevent drying of the cells. Influenza samples were prepared, as described above, 10-fold serially diluted in 0.3% BSA in DMEM and kept on ice. Diluted samples were added to appropriate wells, 100 µl/well, and adsorbed for 1 h at 37 °C, 95% relative humidity, and 5% $CO_2$ while gently rocking. The samples were aspirated and the cells rinsed 1× with PBS + 50 U/ml penicillin + 50 µg/ml + streptomycin. 2× L-15 (Sigma Aldrich) with 25 mM HEPES + 0.15% NaHCO₃ + 100 U/ml penicillin + 100 µg/ml + streptomycin + 0.5 µg/ml gentamicin + 2 µg/ml TPCK-trypsin (Sigma Aldrich) was combined with an equal volume of liquefied 1.0% agarose in water then 2 ml/well was dispensed into the culture wells. After incubating the plates for 2 days at 37 °C, 95% relative humidity, and 5% $CO_2$ the L-15/agar was removed and the cells fixed with 2 ml/well 90% ethanol for 30 min on an orbital shaker at room temperature. The ethanol was replaced with 2 ml/well 0.3% crystal violet (Sigma Aldrich) in 5% isopropanol + 5% ethanol in water and incubated for 20 min at room temperature on an orbital shaker. Finally, the crystal violet was removed, the cells rinsed with 2 ml/well of water, and the cultures allowed to air dry prior to counting the virus plaques.

*Human monocyte-derived macrophage Maraba exposure.* The attenuated strain MG1 Maraba virus has been previously described[30,47]. Maraba virus was prepared and titered at McMaster University, shipped on dry ice to RPCCC and stored at −80 °C[48].

**ELISA.** Cytokines production were measured in supernatants of the cultured cells using ELISA kits for TNF-α, IL-1β, IFN-α, and IL6 (R&D Systems, Minneapolis, MN) following the instructions of the manufacturer.

**Flow cytometry.** The following antibodies were used: anti-human CD33 PE-Cyanine7 (eBioscience, San Diego, CA), anti-human CD86 APC (BioLegend, San Diego, CA), and anti-human CD206 FITC (BioLegend). Annexin V FITC (BioLegend) and propidium iodide solution (Sigma) were used for detection of cell death and apoptosis. Flow cytometry analysis was conducted on a LSRII flow cytometer (Becton Dickinson, Franklin Lakes, NJ). Forward scatter versus side scatter gating was set to include all non-aggregated cells from at least 20,000 events collected per sample and data were analyzed using Winlist 3D version 8.0 (Verity, Topsham ME). CD33+ cells from the total cells were gated to obtain CD33+CD86 + or CD33+CD206+ cell populations based on unstained controls.

**SeaHorse assay.** For all extracellular flux assays, cell were plated on cell-take coated SeaHorse XF96 cell culture microplates at a density of $0.5 \times 10^5$ cells per well. The assay plates were spun for 5 min at 1000 rpm and incubated at 37 °C without $CO_2$ before performing the assay on the SeaHorse Bioscience XFe96. The Mitochondrial Stress Test was conducted in XF media containing 10 mM glucose, 1 mM sodium pyruvate, and 2mM L-glutamine, and the following inhibitors were added at the final concentrations: oligomycin (2 µM), carbonyl cyanide 4-(trifluoromethoxy) phenylhydrazone (FCCP) (2 µM), and rotenone/antimycin A (0.5 µM each). The glycolytic stress test was performed in XF media containing 2 mM L-glutamine, and the following reagents were added at the final concentrations: glucose (10 mM), oligomycin (2 µM), and 2-deoxyglucose (50 mM). The data was analyzed using Wave software.

**TAM isolation**. Ascites was collected at the time of primary surgery from patients with newly diagnosed stage III epithelial ovarian cancer (EOC) under an protocol approved by RPCCCs's IRB. All subjects signed informed consent before surgery. Ascites cells were pelleted by centrifugation (500 g for 10 min at 4 °C). Cells were either used within 24 h of harvest for flow cytometry and functional studies or frozen in liquid nitrogen in media containing 20% FBS and 5% DMSO. PE-Cy7-CD33, PE-CD15 (Beckman Coulter, Brea, CA) and APC-H7-CD14 (Biolegend), anti-human mAb along with L/D Aqua were used for sorting of TAM (CD33+CD15−CD14+) from cryopreserved peritoneal cells and the post-sort purity was >90%. Universal type IFN-1, a hybrid of amino-terminal IFNα-2 and carboxy-terminal IFNα-1, produced in *E. coli*, is used to evaluate RNA editing induction in TAMs and was obtained from PBL Assay Science (Piscataway, NJ).

**RNA sequencing for A3A SC vs. A3A KD transfected macrophages**
*Differential expression analysis of RNA data*. We used a similar approach for RNA_Seq analysis as previously described[23]. Three experimental conditions including M0 macrophages, control SC siRNA, and A3A siRNA (KD) are used. SC and KD samples were derived from the same three donors, whereas the M0 macrophages were derived from three unrelated donors. RNA libraries were prepared using the Illumina TruSeq RNA Exome protocol. For each sample, a total of ~90 million paired-end reads were obtained and QCed using fastqc (v0.10.1)[49]. The reads were mapped to GRCh38 human reference genome and GENCODE (v25) annotation database using TopHat (v2.0.13)[50] with a maximum of 1 mismatch per read. The aligned reads were further checked with RSeQC (v2.6.3)[51] to identify potential RNASeq-related problems. Gene level read counts were estimated with featureCounts from Subread (v1.6.0)[52] using—fracOverlap 1 option. Differential expression analyses were performed using DESeq2 (v1.18.1)[53] and the heatmaps were generated using pheatmap (v1.0.8) R library[54]. Pathways analyses were performed running GSEA (v3.0b3)[55] pre-ranked mode with genes ordered by DESeq2's test statistic and MiSigDB canonical curated gene sets (c2).

*RNA editing detection*. RNA reads were mapped to reference genome GRCh38 using a lenient alignment strategy allowing at most two mismatches per read. Each mapped sample was run through the GATK best practices pipeline to obtain a list of variant sites. A filtered list of candidate editing events was collected based on selecting only variants that match the editing event of interest (i.e. C>T along with G>A and A>G along with T>C). Returning to the mapping results, samples are 'piledup' on the union of all potential candidates. Piledup samples were formatted and compiled for statistical comparisons based on a series of specific conditional filtering. Potential candidates for RNA editing required the following features: (A) variant spots with at least two piledup records per group (SC, KD, or M0) and (B) at least two samples from the experimental condition SC have >5% of editing level (el). A generalized linear model (GLM) was used to model and call editing events within the populations. The number of alternative nucleotides was compared to the reference between the two groups (Group). The first filter was designed to remove any possible known SNPs. The resulting editing candidates were annotated according to dbSNP144. A first layer of filtering removed those events where at least one sample had >0.95 event level, corresponding to homozygous SNPs. Then a second layer was performed where an event was removed if all of the samples had any of the following SNP features: el < 0.05; 0.4 < el < 0.6 and el > 0.95 and if this event was matched in dbSNP database. The next filtering step removed all C>T (or G>A) events that were not preceded by C or T in the +strand and G or A in the −strand. A resulting table specifying the editing site, the type of editing event, editing level and number of reads on a reference and alternative bases on each sample for each group was generated. The editing events were annotated using ANNOVAR's gene-based annotation to identify gene features and protein amino acid sequence changes. Finally, a manual stringency filter was employed to retain only the exonic or UTR sites where (a) the editing level increases 2-fold or more in SC relative to M0 and (b) the edited C is located at the 3'-end of a putative tri- or tetra loop that is flanked by a stem that has at least 2 bp long perfect complementarity, or at least 4 bp long imperfect complementarity in which 1 nucleotide mismatch or bulging was allowed[24]. Sequence data from RNA_Seq experiments are deposited to GEO repository at https://www.ncbi.nlm.nih.gov/geo/ under accession number GSE146867.

**Sanger sequencing**. PCR-amplified cDNA fragments were sequenced to examine candidate RNA-editing sites. PCR reactions were treated with an ExoSAP-IT exonuclease (Affymetrix, Santa Clara, CA) and then directly used for sequencing on 3130 XL Genetic Analyzer (Life Technologies) via RPCCC Genomic Core Facility. Major and minor chromatogram peak heights at a nucleotide position of interest were visualized using Sequencher 5.0 software (Gene Codes, Ann Arbor, MI).

**Statistics and reproducibility**
*Sample size estimation*. No explicit power analysis is used for sample size estimation. We tested whether APOBEC3A has a large effect size on inflammatory marker secretion/expression. The following points, not provided in the manuscript, were considered to enhance the reproducibility of our RNA-Seq findings. The sample numbers in the experiments were limited by the availability of donors.

*RNA_Seq experiments*. Three paired biological replicates for M1 macrophages, each transfected by either SC control or APOBEC3A KD siRNAs, are used to compare gene expression and RNA-editing patterns. Another three biological replicate was used for M0 macrophages. To confidently detect an RNA editing event in a single sample with 5% editing level, we need a coverage of 106 reads at that site with a minimum of three reads with editing. Given that the approximate size of all human transcripts is 50 Mbp, using 100 bp single end reads, we will need 53 million reads mapped to transcriptome. Since RNASeq is different from exome sequencing, in that it depends heavily on the expression level, a higher number of reads (>50 million) is thus desired. In these experiments, we used ~90 million paired-end reads. Stringent C>U RNA editing detection filters included (a) removal of genomic SNPs, (b) removal of C>U sites that are not preceded by a C or T at −1 location, (c) at least 2-fold increase in M1 SC vs. M0 in a candidate editing site, and (d) removal of edited sites that are not located within a putative stem-loop structure. A fourth biological replicate is used to confirm selected RNA editing by Sanger Sequencing (Fig. 4B).

*Replicates*. Each experiment reporting novel findings is performed using independent biological replicates. This information is included in figure legends. Sequence data from RNA_Seq experiments are deposited to GEO repository at https://www.ncbi.nlm.nih.gov/geo/ under accession number GSE146867 (publicly available).

*Statistical reporting*. Statistical methods used are explained in figure legends. We used two-tailed paired *t*-test in experiments comparing the impact of APOBEC3A KD on a cellular phenotype. No data point was excluded from analyses. We employed two normality tests on paired samples: Shapiro–Wilk and Kolmogorov–Smirnov with Dallal–Wilkinson–Liliefor *P* value, as implemented in GraphPad Prism 7.03 (San Diego, CA). The *N* values were too small to run D'Agostino and Pearson normality test. Most data passed both normality tests, and we ran paired *t*-test on them. This is consistent with general ad hoc rule of thumb that the paired *t*-test (unlike the unpaired *t* test) generally meets the approximate normality assumptions in that differences of random variables tend to be more symmetrical even if the raw observations are asymmetrical. Unpaired data are compared by non-parametric Mann–Whitney test. All reported *P*-values are two-tailed. Data are displayed in scatter plot by bars including standard deviation.

**Reporting summary**. Further information on research design is available in the Nature Research Reporting Summary linked to this article.

## Data availability
Sequence data from RNA_Seq experiments are deposited to GEO repository at https://www.ncbi.nlm.nih.gov/geo/ under accession number GSE146867. Raw data points used in Figs. 1, 2, 6, 7, and Supplementary Figs. 2 and 6 are provided in Supplementary Data 3.

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

## Acknowledgements

This research was supported by educational funds from Saudi Arabia (EAQ), startup funds from the Departments of Pathology (B.E.B.), and NIH grant R01CA188900 (B.H.S.) and by National Cancer Institute (NCI) Grant (P30CA016056) involving the use of RPCCC's Genomics Shared Resources, Bioinformatics Shared Resources, Flow Cytometry and Imaging, Immune Analysis Facilities and Clinical Research Services. J.W. and E.C.G. are also supported by U24CA232979. B.A.D. is supported by NIH grants R01HL151498 and R41AI149954.

## Author contributions

B.E.B. and B.H.S. conceived the study and designed the experiments with contributions from E.Y.A. E.Y.A. performed most of the experiments with contributions from S.S. (RNA editing), A.N.H.K. (cell purification and culture), T.E. (cell purification and culture), K.L.S. (cell purification and culture), A.A. (Western blot experiments), B.A.D. (Influenza virus growth and maintenance), Q.L. (statistical support) and J.M., A.J.R.M., and B.D.L. (Maraba virus growth and maintenance) with support from K.O., K.B.M., and K.O. contributed to experimental planning. E.Y.A., B.H.S., and B.E.B. wrote the manuscript. E.Y.A. and B.E.B. prepared the figures. E.C.G. and J.W. analyzed the RNA-seq data and wrote the method. All authors read and approved the final manuscript.

## Competing interests

The authors declare no competing interests.
