## [Peer Review File · Communications Biology]

Reviewers' comments:

Reviewer #1 (Remarks to the Author):

The manuscript "RNA editing enzyme APOBEC3A promotes pro-inflammatory (M1) macrophage polarization by Alqassim et al is probably the first study that found a possible functional significance for RNA editing by APOBEC3A.

The authors give a convincing evidence that APOBEC3a editing may be involved in polarization of M1 macrophages.

The manuscript is read well and the results are sound. Yet, there are few issues that can increase the validity of the results.

1- APOBEC3A is a known DNA editing enzyme, thus it is not fully clear from the manuscript how the authors exclude the option that the editing events that are observed here are not DNA editing events at sub-population of the cells but rather RNA events.

2- The RNA editing detection process is described in details but lack a critical control- the authors should perform the exact same analysis for all types of mismatches (e.g A-C, A-G..) to demonstrate that how sensitive their computational pipeline is.

3- Key manuscripts about APOBEC3A as DNA editors are missing (e.g Buisson et al 2019)

Reviewer #2 (Remarks to the Author):

The paper by Baysal and colleagues does contain information that is scientifically interesting. However, it requires further work and substantial correction before it can be considered for re-review by any journal. Some examples are listed below – but please bear in mind that the issues are not limited to these specific examples.

(1) Multiple instances of statements without supporting citation or evidence; also multiple instances of potentially misleading use of words, for example:

Line 191-193, the authors refer to an editing event in THOC as deleterious without providing a citation for this or providing convincing evidence to support why they have made this statement.

Lines 199-200: The word 'mutation' is incorrectly used to refer to an RNA editing event.

Lines 253-254, the authors claim that the most likely reason for what they have seen is RNA editing by APOBEC3A, where they have only shown expression data that might be linked to this.

(2) All this could be excused as a language problem (and there are many instances of grammatical errors, sentences with missing words or repetitive sentences (see lines 42-43, 72, 114-17 as examples). However, there are also substantial inconsistencies in the presentation and analysis of the data, pointing to thinking process errors vs simple grammar problems. For example:

- Figures should depict data as scatter plots and not bar charts.

- Standard error of the mean should be replaced by standard deviation which is more appropriate and provides a more accurate representation of the data.

- Consistency of the information displayed on an axis should be maintained when showing the same data e.g figure 6E one part of the figure displays MFI units on the Y axis and the other states %CD206.

–The authors have also failed to indicate in their methods how RNA editing percentage was calculated when the data did not originate from NGS RNA sequencing.

Reviewer #3 (Remarks to the Author):

This work looks at the role of the RNA editing enzyme on the inflammatory polarisation of macrophages. It focuses on the editing enzyme APOBEC3A and identifies mRNAs that this enzyme targets in human macrophages. It also shows that knockdown of this enzyme decreases pro-inflammatory cytokine production. While potentially interesting, the current study is too preliminary and does not identify the mechanism by which APOBEC3A polarises macrophages. The initial focus is on succinate dehydrogenase, an enzyme implicated in the metabolic shift that occurs on M1 macrophage polarisation. They show modification of SDHB mRNA, but do not examine if this has an effect on the level of SDHB protein in the cell. The metabolic assays performed indicate an increase in OxPhos suggesting that this could not explain the effects of APOBEC3A knockdown on macrophage polarisation.

In addition, there are several technical issues.

- 1) Controls need to be included to show knockdown efficiency, ideally at the protein level, for APOBEC3A knockdown. The authors should also check the siRNA used to not induce interferon in the cells.
- 2) Does the RNA editing in SDHB affect its protein level?
- 3) Statistical testing – the authors run several different tests. While they explain their rationale, it is worth noting that much of their data is in the form of ratios and percentages and so the authors should consider transformation of the data. The low n number will make normality testing problematic. In their data table they list n=5 for 2A; this is incorrect as n=3 or 2 depending on the virus.
- 4) 'Biological replicate' should be more clearly defined – does this always mean experiments from different donors?
- 5) Does infection with influenza also increase interferon? Do the infections affect cell viability?
- 6) In Fig 3 and 5 Volcano plots would provide a good representation of changes and significance in the complete data set.
- 7) The traces for the Seahorse experiments should be shown, rather than a subset of calculated values.
- 8) Complete experimentation information, such as times and concentrations should be included in the legends.
- 9) The significance of the 2 colours in the FACS plots (SF1) should be stated.
- 10) Accession numbers for the RNAseq data should be clearly stated in the methods.
- 11) There are a number of typos in the manuscript

We thank all 3 reviewers for their constructive comments, which we address them below.

Reviewers' comments:

Reviewer #1 (Remarks to the Author):

The manuscript “RNA editing enzyme APOBEC3A promotes pro-inflammatory (M1) macrophage polarization by Alqassim et al is probably the first study that found a possible functional significance for RNA editing by APOBEC3A.

The authors give a convincing evidence that APOBEC3a editing may be involved in polarization of M1 macrophages.

The manuscript is read well and the results are sound. Yet, there are few issues that can increase the validity of the results.

1- APOBEC3A is a known DNA editing enzyme, thus it is not fully clear from the manuscript how the authors exclude the option that the editing events that are observed here are not DNA editing events at sub-population of the cells but rather RNA events.

In the revised Discussion, we added the following points. “A3A-mediated DNA mutations were previously identified in human cancer cells. We previously examined genomic DNAs corresponding to 23 C>U RNA editing sites in primary human monocytes treated with hypoxia/IFN1 (n=2 donors) and in 293T/APOBEC3A overexpression system (n=2 replicates)²². Separately, we cloned PCR- amplified exon 2 of SDHB genomic DNA, which contains the RNA editing site, in 293T/APOBEC3A overexpression system and sequenced individual clones. In both of these studies, we found no evidence of DNA mutations at sites corresponding to APOBEC3A-induced RNA editing sites. Therefore, we consider cellular RNA editing as the principle mechanism for A3A modulating macrophage function.”

2- The RNA editing detection process is described in details but lack a critical control- the authors should perform the exact same analysis for all types of mismatches (e.g A-C, A-G..) to demonstrate that how sensitive their computational pipeline is.

This analysis is now performed and presented in Supp. Fig. 3. Although RNA mismatches increase in other nucleotide categories as well in SC M1 macrophages compared to M0, the KD of A3A causes the largest declines in CtT and GtA categories, the superset of C>U RNA editing events. This finding is consistent with our hypothesis that A3A mediates widespread C>U RNA editing in M1 macrophages and that the loss of this RNA editing impairs M1 macrophage phenotypes as described in the manuscript.

3- Key manuscripts about APOBEC3A as DNA editors are missing (e.g Buisson et al 2019).

Two references that show DNA editing by APOBEC3A in cancer cells are included. Buisson et al 2019 and Chan et al 2015.

Reviewer #2 (Remarks to the Author):

The paper by Baysal and colleagues does contain information that is scientifically interesting. However, it requires further work and substantial correction before it can be considered for re-

review by any journal. Some examples are listed below – but please bear in mind that the issues are not limited to these specific examples.

(1) Multiple instances of statements without supporting citation or evidence; also multiple instances of potentially misleading use of words, for example:

Line 191-193, the authors refer to an editing event in THOC as deleterious without providing a citation for this or providing convincing evidence to support why they have made this statement. We removed the “deleterious” from Abstract and replaced it with “a highly conserved amino acid”. We now provide additional evidence suggesting the deleterious nature of c.C2047T:p.R683C.

See pages 8 and 9: “THOC5 RNA editing (c.C2047T:p.R683C) alters the last amino acid arginine, which is conserved in all sequences from 100 vertebrate genomes (Supp. Figure 4A). In addition, the edited C and the surrounding nucleotides that predict a stem-loop RNA structure are also highly conserved, including the -1 position, which is either T or C (Supp. Figure 4B). Bioinformatic analyses by PROVEAN, SIFT and POLYPHEN-2 methods respectively predict “deleterious”, “damaging” and “probably damaging” effects of the R683C protein variation. Western blot analysis of M1 SC control and M1 A3A KD samples shows reduced expression of A3A protein by siRNA KD, but similar amounts of THOC5 protein, suggesting that missense alteration by RNA editing does not significantly alter THOC5 protein stability (Figure 4A).”

Lines 199-200: The word ‘mutation’ is incorrectly used to refer to an RNA editing event. The mutation is changed to “altered” in this sentence.

Lines 253-254, the authors claim that the most likely reason for what they have seen is RNA editing by APOBEC3A, where they have only shown expression data that might be linked to this.

We now provide new Western blot results that show reduction in APOBEC3A protein by KD (Fig. 4A), which is consistent with reduction in APOBEC3A gene expression. APOBEC3A KD coordinately reduces RNA editing levels of scores of genes, including *SDHB*. *SDHB* c.136C>U site-specific RNA editing can also be induced in vitro in test tube using recombinant APOBEC3A protein and synthetic RNA (Sharma et al. 2015). As we detailed in response to reviewer#1’s 1st comment and in Discussion, there is no evidence that APOBEC3A is localized to the nucleus or mutates the genomic DNA under conditions that induce RNA editing in *normal* monocytes/macrophages. These considerations and our current data support the conclusion that RNA editing by APOBEC3A is the most likely mechanism that promotes M1 polarization.

(2) All this could be excused as a language problem (and there are many instances of grammatical errors, sentences with missing words or repetitive sentences (see lines 42-43, 72, 114-17 as examples).

The second sentence of Introduction (lines 42-43) is deleted. On line 72, “compare” is changed to “compared”. On line 114-117, 1 of 2 sentences is deleted. It now reads “In this study, we hypothesized that A3A plays an essential role in macrophage functions during M1 polarization and in response to viral infections.” (page 5)

However, there are also substantial inconsistencies in the presentation and analysis of the data, pointing to thinking process errors vs simple grammar problems. For example:

- Figures should depict data as scatter plots and not bar charts.

All Bar plots are now converted to scatter/bar plots which show individual data points.

- Standard error of the mean should be replaced by standard deviation which is more appropriate and provides a more accurate representation of the data. Data dispersion is now shown by SD in figures.

- Consistency of the information displayed on an axis should be maintained when showing the same data e.g figure 6E one part of the figure displays MFI units on the Y axis and the other states %CD206.

We now present data for CD206 as MFI and show a representative flow cytometry trace and the summary data in supplementary figure 6.

–The authors have also failed to indicate in their methods how RNA editing percentage was calculated when the data did not originate from NGS RNA sequencing.

RNA editing percentages were derived from NGS. Please see ‘RNA editing detection’ on page 20.

Reviewer #3 (Remarks to the Author):

This work looks at the role of the RNA editing enzyme on the inflammatory polarisation of macrophages. It focuses on the editing enzyme APOBEC3A and identifies mRNAs that this enzyme targets in human macrophages. It also shows that knockdown of this enzyme decreases pro-inflammatory cytokine production. While potentially interesting, the current study is too preliminary and does not identify the mechanism by which APOBEC3A polarises macrophages. The initial focus is on succinate dehydrogenase, an enzyme implicated in the metabolic shift that occurs on M1 macrophage polarisation. They show modification of SDHB mRNA, but do not examine if this has an effect on the level of SDHB protein in the cell. The metabolic assays performed indicate an increase in OxPhos suggesting that this could not explain the effects of APOBEC3A knockdown on macrophage polarisation.

Our data suggest that transcriptome-wide RNA editing is the mechanism by which APOBEC3A facilitates M1 polarization. Whether high-levels of RNA editing of a few genes (e.g. *THOC5*) or most or the totality of RNA editing events is responsible for this function is the subject of future studies. We used *SDHB* variant-specific RT-qPCR to test the induction or loss of the RNA editing function by APOBEC3A. *SDHB* is only one of dozens of genes that are targeted by APOBEC3-mediated RNA editing. As shown in Figure 7, we find no evidence that APOBEC3A contributes to glycolytic switch or to basal oxygen consumption in M1 macrophages. As part of this revision, we also performed *SDHB* western blot analysis and find no appreciable differences in protein quantity between SC control and APOBEC3A KD (Fig. 4A). This is not unexpected since *SDHB* R46X nonsense RNA editing levels are ~15% in M1 macrophages (Fig. 1B).

In addition, there are several technical issues.

1) Controls need to be included to show knockdown efficiency, ideally at the protein level, for APOBEC3A knockdown. The authors should also check the siRNA used to not induce interferon in the cells.

APOBEC3A KD efficiency using our protocol is demonstrated by Western blot (Fig. 4A). We now include data showing that SC control or A3A KD siRNAs do not induce IFN-alpha in M0 macrophages (Fig. 2B). Consistent with this finding, *SDHB* RNA editing levels in SC siRNA-transfected M0- and M2-polarized macrophages are insignificant (<5%, Fig. 1A).

2) Does the RNA editing in *SDHB* affect its protein level?

We find no appreciable differences in *SDHB* protein quantity between M1 A3A SC control and A3A KD by Western blot analysis. In M1 A3A SC samples the c.136C>U (R46X) RNA editing levels are ~15% (Fig. 1B), which is not predicted to dramatically reduce protein levels.

3) Statistical testing – the authors run several different tests. While they explain their rationale, it is worth noting that much of their data is in the form of ratios and percentages and so the authors should consider transformation of the data.

The low n number will make normality testing problematic.

We have now incorporated at the end of Methods “Statistics and Reproducibility”, explaining our statistical approaches: “We employed two normality tests on paired samples: Shapiro-Wilk and Kolmogorov-Smirnov with Dallal-Wilkinson-Liliefors P value, as implemented in GraphPad Prism 7.03. The N values were too small to run D’Agostino & Pearson normality test. Most data passed both normality tests, and we ran paired t-test on them. This is consistent with general ad hoc rule of thumb that the paired t-test (unlike the unpaired t test) generally meets the approximate normality assumptions in that differences of random variables tend to be more symmetrical even if the raw observations are asymmetrical. Unpaired data are compared by non-parametric Mann-Whitney test. All reported p-values are two-tailed. Data are displayed in scatter plot by bars including standard deviation.”

In their data table they list n=5 for 2A; this is incorrect as n=3 or 2 depending on the virus.

In Fig. 2A, we test the null hypothesis that virus infection (*Maraba or Influenza*) does not induce A3A-mediated RNA editing. We combine data from both viruses for this test, and individual data points are displayed for each virus.

4) ‘Biological replicate’ should be more clearly defined – does this always mean experiments from different donors?

Yes. All experiments are from different donors (biological replicate), unless otherwise specified.

5) Does infection with influenza also increase interferon? Do the infections affect cell viability?

We include new data showing influenza virus infection increases IFN-alpha levels and decreases cell viability (see Fig. 2B and Supp. Fig.2).

6) In Fig 3 and 5 Volcano plots would provide a good representation of changes and significance in the complete data set.

Differential gene expression between control SC M1 and A3A KD M1 is now shown as a volcano plot (see new Fig. 5). Differential RNA editing data between M0, control SC M1 and

APOBEC3A KD M1 (Fig.3) are kept the same, which we believe that it provides better visualization of RNA editing changes.

7) The traces for the seahorse experiments should be shown, rather than a subset of calculated values.

We show seahorse traces for 1 experiment in new figure 7. The rest of the data files is deposited.

8) Complete experimentation information, such as times and concentrations should be included in the legends.

Times and concentrations are included in legends.

9) The significance of the 2 colours in the FACS plots (SF1) should be stated.

The figure is now in uniform color.

10) Accession numbers for the RNAseq data should be clearly stated in the methods.

Done (p. 21).

11) There are a number of typos in the manuscript.

The manuscript is proofread to eliminate typos.

REVIEWERS' COMMENTS:

Reviewer #1 (Remarks to the Author):

The Authors addressed my concerns and i believe the manuscript can be published.

Reviewer #3 (Remarks to the Author):

The authors have answered the technical comments I rasied, and the manuscript is improved. The molecular mechanism by which APOBEC3A affcets macrophage polarisation is not resloved, although I accept that with the potentila editing of multiple mRNAs this may be hard to tie down.